genomics/evolution/ecology

Ziphiidae, mitogenomes, nuclear genomes, population structure, evolution, demographic history

**Author for correspondence:**
K. F. Thompson
e-mail: k.f.thompson@exeter.ac.uk

†These authors contributed equally to this study.

# Ocean-wide genomic variation in Gray's beaked whales, *Mesoplodon grayi*

M. V. Westbury[1,†], K. F. Thompson[2,†], M. Louis[1], A. A. Cabrera[1], M. Skovrind[1], J. A. S. Castruita[1], R. Constantine[3], J. R. Stevens[2] and E. D. Lorenzen[1]

[1]GLOBE Institute, University of Copenhagen, Denmark
[2]Biosciences, University of Exeter, Devon, UK
[3]School of Biological Sciences and Institute of Marine Science, University of Auckland, New Zealand

MVW, 0000-0003-0478-3930; KFT, 0000-0003-4277-3549;
ML, 0000-0002-4611-5503; AAC, 0000-0001-5385-1114;
RC, 0000-0003-3260-539X; JRS, 0000-0002-1317-6721;
EDL, 0000-0002-6353-2819

The deep oceans of the Southern Hemisphere are home to several elusive and poorly studied marine megafauna. In the absence of robust observational data for these species, genetic data can aid inferences on population connectivity, demography and ecology. A previous investigation of genetic diversity and population structure in Gray's beaked whale (*Mesoplodon grayi*) from Western Australia and New Zealand found high levels of mtDNA diversity, no geographic structure and stable demographic history. To further investigate phylogeographic and demographic patterns across their range, we generated complete mitochondrial and partial nuclear genomes of 16 of the individuals previously analysed and included additional samples from South Africa ($n = 2$) and South Australia ($n = 4$), greatly expanding the spatial range of genomic data for the species. Gray's beaked whales are highly elusive and rarely observed, and our data represents a unique and geographically broad dataset. We find relatively high levels of diversity in the mitochondrial genome, despite an absence of population structure at the mitochondrial and nuclear level. Demographic analyses suggest these whales existed at stable levels over at least the past 1.1 million years, with an approximately twofold increase in female effective population size approximately 250 thousand years ago, coinciding with a period of increased Southern Ocean productivity, sea surface temperature and a potential expansion of suitable habitat. Our results suggest that Gray's beaked whales are likely to be resilient to near-future ecosystem changes, facilitating their

conservation. Our study demonstrates the utility of low-effort shotgun sequencing in providing ecological information on highly elusive species.

## 1. Introduction

The ocean is undergoing rapid ecological change as a result of climate change, ocean acidification and deoxygenation, poor fisheries management, oil and gas extraction, litter and shipping [1]. In an attempt to protect biodiversity, the United Nations is negotiating a treaty to ensure the conservation and sustainable use of biodiversity in the areas beyond jurisdiction—the 'high seas'—or the deep ocean beyond the continental shelf edges (https://www.un.org/bbnj/). However, despite the desire to protect biodiversity in the high seas, the task is a challenge when the ecology of elusive species is still a mystery.

Currently, the deep sea is one of the last frontiers of exploration on Earth and there are a great many species for which we know very little, including large cetaceans [2,3]. Despite their general public appeal and influence on human culture throughout the years [4], oceanic cetaceans, including beaked whales, dolphins and baleen whales, are generally poorly understood. For the beaked whales, basic data on migrations, foraging ecology and social structures are lacking for most species, leading to difficulties in assessing how they will respond as their key habitats change, which leads to the question of how vulnerable they may be to broad marine ecosystem changes.

The taxonomy of oceanic beaked whales has been substantially revised during the past decade, mainly through the use of genetic tools, and there is much to learn about this family of deep-water cetaceans. Recent examples of newly described or taxonomically resurrected beaked whale species include: spade-toothed whale (*Mesoplodon traversii*) in 2012, Derinayagala's whale (*Mesoplodon hotaula*) in 2014 and a new species of Baird's beaked whale (*Berardius minimus*) in 2019 [5–7].

There are 23 currently recognized species of beaked whales within five genera [8]. All species are deep divers that feed on squid, small fish and, for some, crustaceans [9,10]. These whales have the most extreme diving behaviour of all marine mammals, with some species diving to nearly 3000 m deep, and remaining submerged for up to three hours (Cuvier's beaked whales (*Ziphius cavirostris*) [11]). Beaked whales inhabit areas off the continental shelf edge out to the deep waters. This habitat preference, alongside their elusive surface behaviour, means that only a few species and populations have been well-studied. In general, populations around oceanic islands are better understood [12–16]. Beaked whales that have an oceanic distribution far offshore in the Southern Hemisphere are very rarely observed alive, let alone researched, limiting insights into their ecology [12].

Gray's beaked whales (*Mesoplodon grayi*) are medium-sized (approx. 5 m) whales, with a Southern Hemisphere, circumpolar distribution south of 33° latitude (figure 1) [20]. There are no estimates of the abundance of Gray's beaked whales based on census data. The use of genetic data has provided ecological insights into this species, which could not be achieved with more conventional methods due to their elusive nature. Previous work using partial mtDNA control region sequences (530 bp) and 12 microsatellites from 94 Gray's stranded on the coast of Western Australia and New Zealand uncovered relatively high levels of genetic diversity, and no population differentiation between localities [20]. The study estimated a female effective population size of 460 000 whales, albeit with very large confidence intervals (95% CI 20 000–2 250 000). Moreover, investigations into the genetic kinships within different stranded groups in Western Australia and New Zealand found no indications of related adults stranding together, suggesting both sexes disperse from their natal group to potentially form groups with unrelated individuals [21]. Such a flexible social system combined with the rich oceanic habitat of the Southern Hemisphere could facilitate gene flow and potentially support a surprisingly large number of Gray's beaked whales.

Here, we present a new reference panel of complete mitochondrial genomes and partial nuclear genomes to test the hypothesis of no clear phylogeographic structuring in the elusive Gray's beaked whale. Our study builds on previous work through the inclusion of genomic sequencing and an increased geographic range of samples by including an additional two samples from South Africa and four samples from South Australia. Despite the increase in genetic data, and the inclusion of individuals from outside the previous study range, we find a lack of population structure to hold for the wider distribution of the species. Moreover, we provide new insights into the maternal demographic history and genetic diversity of Gray's beaked whales and find the species has existed at relatively high population densities for at least the past one million years.

(a)

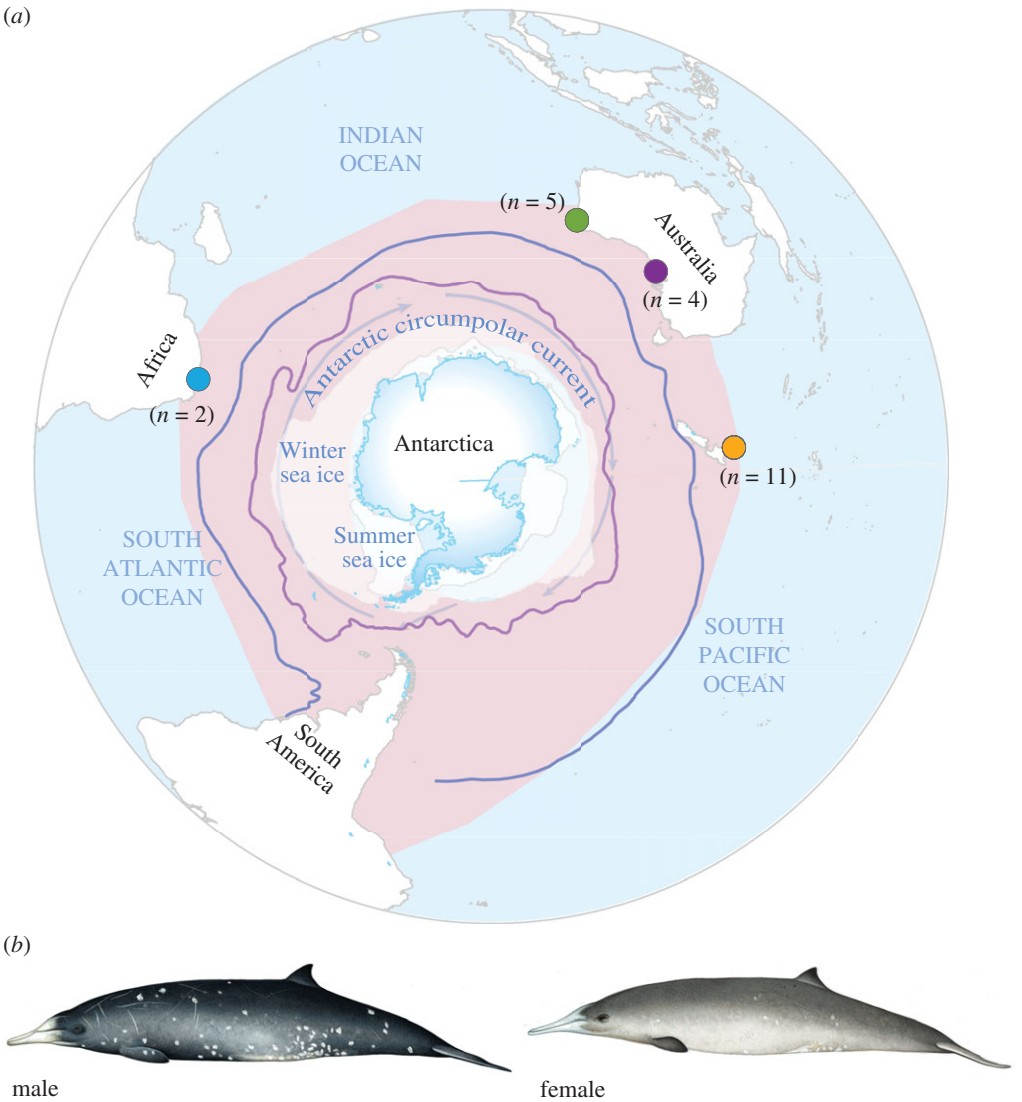

(b)

male                                              female

**Figure 1.** Distribution, sampling locations and morphology of Gray's beaked whale. (*a*) Pink shaded area represents species distribution based on the IUCN Red List of Threatened Species [17]. Coloured circles denote sample locations: South Africa (blue), Western Australia (green), South Australia (purple), New Zealand (orange), and numbers denote sample sizes. Winter and summer sea ice coverage are indicated by white shading and correspond to average sea ice coverage during the months of March and September 2016 [18]. Subtropical convergence (blue line) and Antarctic polar front (purple line) are indicated [19]. Map scale is 1 : 95 000 000. (*b*) Male and female Gray's beaked whales; illustration by Mark Camm.

## 2. Material and methods

### 2.1. Samples

We analysed a total of 22 samples from South Africa ($n = 2$), Western Australia ($n = 5$), South Australia (n = 4) and New Zealand ($n = 11$) (figure 1). The samples from Western Australia and New Zealand were included in previous genetic analyses [20]. Samples were collected from beach cast whales during 1999–2013 and provided by the Port Elizabeth Museum (South Africa), South Australian Museum and New Zealand Cetacean Tissue Archive (electronic supplementary material, table S1).

We mapped the distribution of the sampling locations in ArcGIS® (v. 10.3, Esri® Inc.), using a South Pole Lambert azimuthal equal area projection and a World Geodetic System 1984 with map datum at a scale of 1 : 95 000 000, and included the most recent projection of Gray's beaked whale range (figure 1) [20]. The map includes environmental data within Gray's beaked whale habitat; to identify contemporary sea ice extents, ice sheets and glacial projections, we used data from the National Snow and Ice Data Center [18] from Natural Earth 2017. Data from Orsi & Harris [19] were used to identify the location of the subtropical convergence and Antarctic polar front.

## 2.2. Laboratory analyses

Using the 22 samples available from an approximately 13 000 km range of Gray's beaked whale distribution, we set out to sequence both complete mitochondrial genomes and partial nuclear genomes from shotgun sequencing data. We extracted DNA using a DNeasy blood and tissue kit (Qiagen) following the manufacturer's protocol and sonicated to approximately 450 bp using a Covaris. From the sonicated DNA, we built Illumina sequencing libraries using the BEST protocol [22], using an Illumina adapter mix concentration of 20 μM and 15 cycles during the indexing PCR step. Indexed libraries were cleaned using a SPRI bead DNA purification method. All samples were multiplexed in approximately equimolar amounts and sequenced on an Illumina Hiseq 4000 at the GeoGenetics Sequencing Core, University of Copenhagen, using 80 bp single-end (SE) and 150 bp paired-end (PE) reads.

## 2.3. Mitochondrial DNA analyses

### 2.3.1. Mitochondrial genome assembly

To construct mitochondrial genomes, we initially sequenced the libraries using 80 cycle SE sequencing. We trimmed adapters from the raw reads and removed reads shorter than 30 bp using skewer v. 0.2.2 [23]. We mapped the resultant reads to a Gray's beaked whale mitochondrial genome (Genbank ID: nC_023830.1) using BWA v. 0.7.15 [24] with the mem algorithm, and default parameters. We parsed the output and removed duplicates and reads of mapping quality less than 30 using SAMtools v. 1.6 [25]. We built consensus sequences using a majority rules approach (-doFasta 2) in ANGSD v. 0.921 [26] only considering bases with a base quality score greater than 25 (-minq 25), reads with a mapping quality score greater than 25 (-minmapq 25), and sites with at least 5 x coverage (-minInddepth 5). The resultant consensus sequences were aligned using MAFFT v. 7.392 and default parameters [27].

### 2.3.2. Diversity statistics

We calculated haplotype diversity ($h$) and nucleotide diversity ($\pi$) in DnaSP v. 6.12.03 [28]. To evaluate the relative level of mitochondrial diversity in Gray's beaked whale, we compared their nucleotide diversity ($\pi$) to estimates from 12 other cetaceans obtained using available population-scale mitogenome data from the literature. We used the values previously calculated in Louis *et al*. [29] and added available data from the Blainville's, Cuvier's and Gervais' beaked whales [30] as well as northern bottlenose whales [31], all sampled from across their ranges. Following Louis *et al*. [29], we estimated nucleotide diversity excluding sites with gaps and missing data only in each pairwise comparison in DnaSP.

### 2.3.3. Haplotype network

We used PopArt v. 1.7 [32] to generate a haplotype neighbour-joining network using a 0.1 reticulation tolerance from the aligned mitochondrial genomes.

### 2.3.4. Phylogeny

We built a time-calibrated phylogenetic tree in BEAST v. 2.6 [33] to estimate the divergence times among our Gray's beaked whales. For this analysis, we separated the protein-coding genes ($n = 13$), tRNAs ($n = 22$), rRNAs ($n = 2$) and the control region based on the published annotation of the reference Gray's beaked whale mitogenome [34] which we aligned using MAFFT [27]. We manually checked the alignments and, when required, edited them to match the reading frame of the protein-coding genes.

We combined the data into six subsets that were (i) first, (ii) second and (iii) third codon positions of all protein-coding regions, (iv) tRNAs, (v) rRNAs and (vi) control region. We set all regions to have the same topology and clock rate, but specified optimal substitution models for each of the six subsets as identified according to PartitionFinder2 [35] (electronic supplementary material, table S2). We specified a coalescent constant population model and applied a strict clock as we assume little heterogeneity in clock rate within Gray's. We set the clock rate as 0.0038 substitutions per site per million years based on that previously found using an odontocete fossil-calibrated phylogenetic tree [29].

We ran the Markov chain Monte Carlo (MCMC) twice with 50 000 000 steps logged every 5000 steps. We combined the tree files and log files of the two runs using LogCombiner v. 2.5.1 [36] with a burnin of 10%. We checked stationarity (ESS values above 200 for all parameters) and convergence by comparing posterior distributions between the two chains using Tracer v. 1.7 [37]. We used TreeAnnotator v. 2.5.1

[38] to create a maximum clade credibility tree with mean node heights and a posterior probability limit of 0.9 based on the combined trees. We visualized the tree using FigTree v. 1.4.3 (http://tree.bio.ed.ac.uk/software/figtree/).

### 2.3.5. Demographic history

We reconstructed the demographic history of Gray's beaked whales using a coalescent Bayesian skyline plot analysis (BSP) in BEAST v. 2.6 [33] using the fully partitioned mitogenome sequences and all 22 sequences from the dated phylogeny. The substitution models for each of the six partitions were defined as in the previous step. We ran two MCMC chains of 50 000 000 states with sampling every 5000 states, with a burn-in of 10%. We examined ESS values in Tracer v. 1.7.1, and compared the posterior distributions of the two chains to ensure convergence of the runs [37].

To compare the results of a coalescent constant population model with the BSP model and determine the best model fit for our data, we ran the analyses again with identical parameters and derived marginal likelihoods for both models. To evaluate the models, we used the Nested Sampling (NS) package from BEAST2 [39] with default settings except for the particleCount parameter which was increased to 50. The marginal likelihoods produced by NS and their standard deviations were used to determine which model was the better fit for our data. If the difference between the marginal likelihoods of two models were larger than $2^*\sqrt{(SD1^*SD1 + SD2^*SD2)}$, where $SD1$ and $SD2$ are the standard deviations of the two models being compared, the model with the larger marginal likelihood was favoured. Finally, we ran an extended BSP in BEAST with identical parameters to previous runs to check whether a constant population model was indicated by the sum(indicators.alltrees) parameter that estimated the number of population change steps. To derive $N_{ef}$ we scaled the population size estimates with a generation time of 23.4 years, based on the average for available beaked whale species provided in Taylor et al. [40].

## 2.4. Nuclear DNA analyses

### 2.4.1. Scaffolding Sowerby's beaked whale assembly

The only available beaked whale assemblies (Mesoplodon bidens (GCA_004027085.1—Sowerby's beaked whale), and Ziphius cavirostris (GCA_004364475.1—Cuvier's beaked whale)) are very fragmented (scaffold N50 of 33.5 kb and 3.6 kb respectively), which could impact certain downstream analyses and hinder our ability to locate sex chromosomes for sexing of the individuals. Therefore, we first scaffolded the existing Sowerby's beaked whale assembly using the orca (Orcinus orca) assembly (Genbank accession: GCA_000331955.2), with a modified version of the methodology set out by cross-species scaffolding [41]. Although the orca is highly divergent from the Gray's beaked whale (approx. 30 Ma [42]), we selected the orca as it is the most closely related species with a high-quality genome assembly available.

We downloaded the raw reads from Sowerby's beaked whale from the European Nucleotide Archive (SRR7704819) and mapped these to the orca genome using the same protocol as mentioned above. We created a consensus sequence using a consensus base call approach with ANGSD, and specifying a minimum read depth of 8 (-minInddepth 8), minimum mapping quality of 25 (-minmapq 25), minimum base quality of 25 (-minq 25) and only considering reads that mapped uniquely to one location (-uniqueonly 1). We constructed in-silico mate-paired libraries with approximately 1, 2, 5, 8, 10, 15 and 20 kb insert sizes from this consensus sequence using seq-scripts (https://github.com/thackl/seq-scripts), specifying a read length of 150 bp and read depth of 30×.

Using these in silico mate-paired libraries, we scaffolded the existing Sowerby's beaked whale assembly using SSPACE v. 2 [43] and default parameters. We closed assembly gaps using the adapter-trimmed Sowerby's beaked whale raw reads and abyss-sealer [44], specifying a kmer size of 51. We removed all scaffolds shorter than 1 kb from the gap-closed assembly. We assessed the final assembly quality in the form of contiguity using QUAST [45], and in the form of gene content using BUSCO v. 3 [46] and both the eukaryotic and mammalian BUSCO databases.

### 2.4.2. Mapping to the orca and Sowerby's beaked whale assemblies

To investigate whether our population genomic analyses may be impacted by choice of reference, we mapped our reads (both SE and PE) to the available orca assembly as well as the Sowerby's beaked whale assembly independently. We mapped the 80 bp SE and 150 bp PE reads independently,

following the same procedure as mentioned above for the mitochondrial genomes, and merged the resultant bam files using SAMtools merge.

### 2.4.3. Population genomics: principal component analysis, admixture proportions, phylogeny, fixation index and D-statistics

We performed principal component analysis (PCA) and admixture proportion analysis on the data mapped to each assembly (both the orca and Sowerby's beaked whale) independently using ANGSD v. 0.921, as ANGSD uses genotype likelihoods, taking uncertainty in genotype calling of low coverage into account, instead of directly calling genotypes. We first performed a PCA and admixture proportion analysis in PCAngsd [47], using a beagle file constructed in ANGSD. To construct the beagle file, we called genotype likelihoods using the GATK algorithm (-GL 2), specified the output as a beagle file (-doGlf 2) and applied the following filters: only include reads with a mapping quality greater than 25 (-minmapQ 25), only include bases with base quality greater than 25 (-minQ 25), only include reads that map to one location uniquely (-uniqueonly 1), a minimum minor allele frequency of 0.05 or greater (-minmaf 0.05), only call a SNP if the $p$-value is greater than $1 \times 10^{-6}$ (-SNP_pval $1 \times 10^{-6}$), infer major and minor alleles from genotype likelihoods (-doMajorMinor 1), only include sites if at least 50% of individuals are covered (-MinInd 11), remove sex scaffolds and scaffolds shorter than 100 kb (-rf), and call allele frequencies based on a fixed major and an unknown minor allele (-doMaf 2). In PCAngsd, we specified default parameters apart from the addition of the admixture proportions (-admix). The resultant covariance matrix was converted into a PCA and visualized using R v. 3.4.3 [48].

We further evaluated whether the inclusion of sites in linkage disequilibrium (LD) may have an impact on downstream results. To do this, we also ran these analyses using the data mapped to the Sowerby's beaked whale assembly and a subset of 70 140 sites determined not to be in LD using ngsLD [49] specifying the parameters –max_kb_dist of 20 and –min_weight 0.5.

To further investigate geographic structure in our data, we built a neighbour-joining tree using the data mapped to the Sowerby's beaked whale. We built a distance matrix in ANGSD using a consensus base call approach (-doIBS 2), and the same parameters as the beagle file construction for the PCA. We converted the distance matrix into a nexus tree file using PHYLIP neighbour [50].

As an additional method to investigate for the presence (or absence) of population structure based on geography, we computed the fixation index ($F_{ST}$) between individuals grouped by location. For this analysis, we only used the data mapped to the Sowerby's beaked whale assembly. We computed a consensus haploid call using ANGSD (-dohaplocall 2) and specified the same filtering parameters as the PCA and admixture proportions above. We converted the output haplo file into a geno file by removing the major allele column to be used as input for a python script available online (https://github.com/simonhmartin/genomics_general/blob/master/popgenWindows.py). For this, we grouped individuals based on geographical origin (i.e. South Africa, West Australia, South Australia and New Zealand), and specified a window size of 100 kb, and a minimum number of sites per window as 500 bp. Significance from 0 was evaluated using a one-sample Wilcoxon signed-rank test in R.

As a final test for population structure based on geographical origin, we performed a D-statistics (ABBA/BABA) test for topology (e.g. [51,52]) on the data mapped to the Sowerby's beaked whale assembly. Although D-statistics are most commonly used to find evidence of gene flow, represented by a high D-score, a high D-score can also be caused by more recent common ancestry brought about by an incorrect predefined topology. Taking the latter into account, we placed individuals into four predefined groups based on sampling location (i.e. South Africa, Western Australia, South Australia and New Zealand) and compared the average D values produced from correct topologies, that is, branches H1 and H2 containing individuals from the same locality, and incorrect topologies, that is, branches H2 and H3 have individuals from the same locality, while H1 is from a different locality. Furthermore, to investigate for evidence of subpopulation structure within localities, we also performed this test with individuals from a single locality placed on the H1, H2 and H3 branches.

We computed D-statistics in ANGSD using a random base call approach (-doAbbaBaba 1) and the following parameters: a minimum mapping and quality score of 25 (-minmapQ 25, -minQ 25), only consider reads that map to one location uniquely (-uniqueonly 1), calculate genotype likelihoods using GATK algorithm (-GL 2), calculate major and minor alleles using genotype likelihoods (-doMajorMinor 1), skip sites that are triallelic (-skiptriallelic 1), calculate D using a block size of 1Mb (-blocksize 1 000 000) and specify the Sowerby's beaked whale as the outgroup. We performed a block jackknife approach to test for the significance of the results (Z score) using the Rscript (jackKnife.R)

available as part of the ANGSD toolsuite. ANGSD computes every possible combination of individuals, which we filtered based on the sampling localities of the individuals. Scripts on how to filter the ANGSD output to test for population structure can be found at https://github.com/Mvwestbury/Dstats-topology-test. The significance of these results, i.e. whether |Z| was greater than 3, was determined for each comparison using a one-sample Wilcoxon signed-rank test in R.

### 2.4.4. Relatedness and sex determination

To assess levels of relatedness between individuals originating from the same geographic location, we used NGSrelate v. 2 [53]. As input for this, we calculated genotype likelihoods of our dataset using ANGSD following similar parameters as for the PCA analysis. However, there were some minor adjustments to produce the required input file for NGSrelate, namely the inclusion of -doGlf 3 which prints the output as a binary likelihood file as opposed to a beagle file. The number of sites used between pairwise comparisons ranged from 302 286 to 651 438.

In this study, we used the low-coverage genomes to identify the sex of new and previously analysed samples. Our method was as follows: we identified scaffolds in the orca and Sowerby's beak whale assemblies putatively belonging to sex chromosomes using SatsumaSynteny v. 2.01 [54], default parameters, and aligning each assembly independently to the cow X (Genbank accession: CM008168.2) and human Y (Genbank accession: NC_000024.10) chromosomes. Following this, we used the sex chromosome to autosome read depth ratio to determine the sex of the individuals in the current study. This was done by comparing the mean read depth across scaffolds putatively originating from the X chromosome, to that of scaffolds putatively originating from the autosomes. Moreover, to remove biases that may occur due to homology between some regions of the X and Y chromosomes, we removed any scaffolds that aligned to both the X and Y chromosomes from this comparison. This resulted in X-linked scaffolds totalling a length of approximately 120 Mb. We calculated the average read depth of these scaffolds using the depth function in SAMtools. We further selected scaffolds totalling approximately 120 Mb that did not align to the X or Y chromosomes and calculated the average read depth along these scaffolds for comparison.

# 3. Results

## 3.1. Mitochondrial DNA

### 3.1.1. Mitogenomes: coverage, diversity and population structure

We constructed mitochondrial genomes from all 22 individuals to coverages of between 27.0× and 112.7×, with a mean coverage of 58.3× (electronic supplementary material, table S5). Gray's beaked whales exhibit relatively high mitochondrial nucleotide diversity ($\pi = 0.00742$) in comparison with the other surveyed odontocete species ($\pi = 0.00094$–$0.00941$) (figure 2). The haplotype network and phylogenetic tree show no apparent pattern of geographic structure across the study area (figure 3a,b).

### 3.1.2. Demographic history

We analysed the demographic structure of the mitochondrial genomes with two different models (BSP and constant population). The marginal likelihoods of the models show that the BSP model is the best fit for our data (marginal likelihood (s.d.), BSP = −26590.39 (1.91) and constant = −26611.62 (2.04)). In addition, the extended BSP analyses indicated that we can confidently reject a constant population since the summ(indicator.alltrees) 95% HPD excluded 0. The BSP analysis suggests that Gray's beaked whales have existed as a relatively stable population over at least the past 1.1 million years (figure 4a) with an approximate doubling of female effective population size ($N_{ef}$) around approximately 250 thousand years ago (kya). The current estimated median $N_{ef}$ is 270 500 whales (95% CI 116 544– 1 244 228). Our findings are broadly consistent with those reported by Thompson *et al.* [20,34]. Our estimate of $N_{ef}$ falls within the range reported in Thompson *et al.* [20,34] that used a mutation rate estimated for Cuvier's beaked whale ($N_{ef}$ 433 339 whales, 95% CI 16 667–2 213 248). Our study indicates a BSP trajectory that is similar to previous work, but the current genomic analyses have been able to estimate $N_{ef}$ and the timing of population expansions with greater confidence and more resolution.

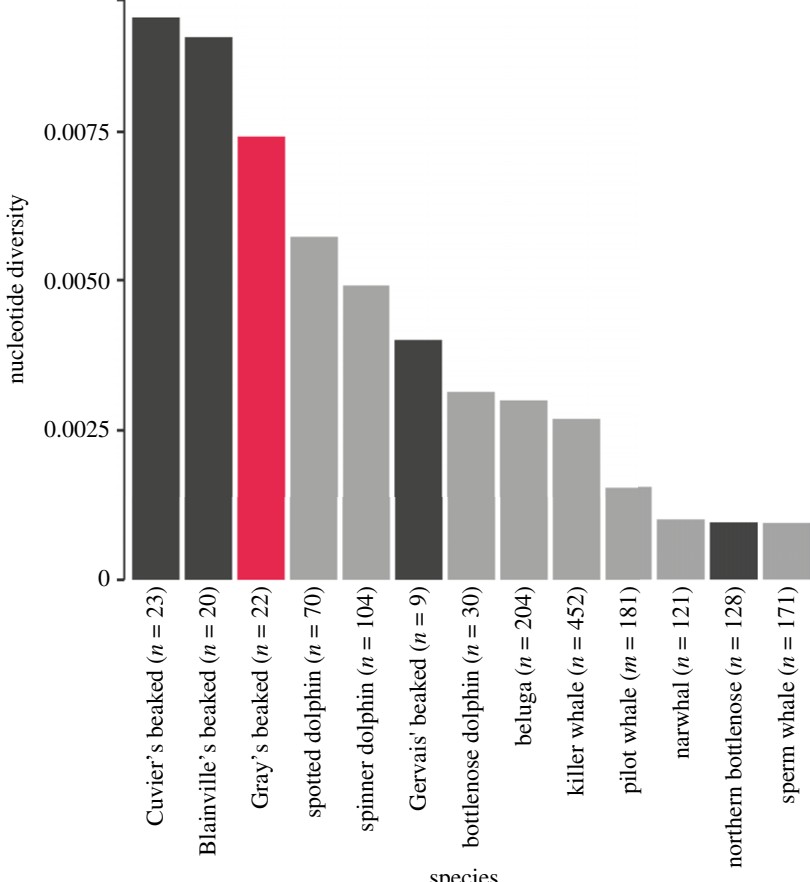

**Figure 2.** Comparison of nucleotide diversity ($\pi$) derived from population-level studies of 13 toothed whale mitochondrial genomes. Gray's beaked whales are shown in red, and other beaked whales are shown in dark grey. Sample sizes for each species are given in the legend.

## 3.2. Nuclear DNA

### 3.2.1. Newly scaffolded Sowerby's beaked whale assembly

We scaffolded the publicly available Sowerby's beaked whale contig-level genome assembly (N50 33.5 kb) using *in silico* mate-paired libraries of between 1 and 20 kb in length and SSPACE v. 2 [43], resulting in an assembly of 2 356 367 458 bp in total, with a scaffold N50 of 2.6 Mb (electronic supplementary material, table S3) and a mammalian database BUSCO score of 93% complete BUSCOs (electronic supplementary material, table S4). From this assembly, we successfully identified 145 scaffolds possibly associated with the X chromosome to be used for downstream sex determination analyses.

### 3.2.2. Nuclear mapping results

We mapped our data independently to two nuclear reference genomes. There was an average of approximately 13% less bp mapping to the orca assembly relative to the Sowerby's beaked whale assembly (tables S5 and S6). The higher mapping count when mapping to the more closely related Sowerby's beaked whale genome is not unexpected, as the orca diverged from the Gray's beaked whale approximately 29.98 Ma (CI 29.0–30.94), while Sowerby's beaked whale diverged only approximately 7.42 Ma (CI 6.67–8.18) [42].

### 3.2.3. Population genomics: principal component analysis, admixture proportions, phylogeny, fixation index and D-statistics

To investigate the biparental population structure within the Gray's beaked whales, we applied several different population genomic analyses. PCA and admixture proportion analyses separated four New Zealand individuals regardless of the mapping reference and whether a subset of loci determined not

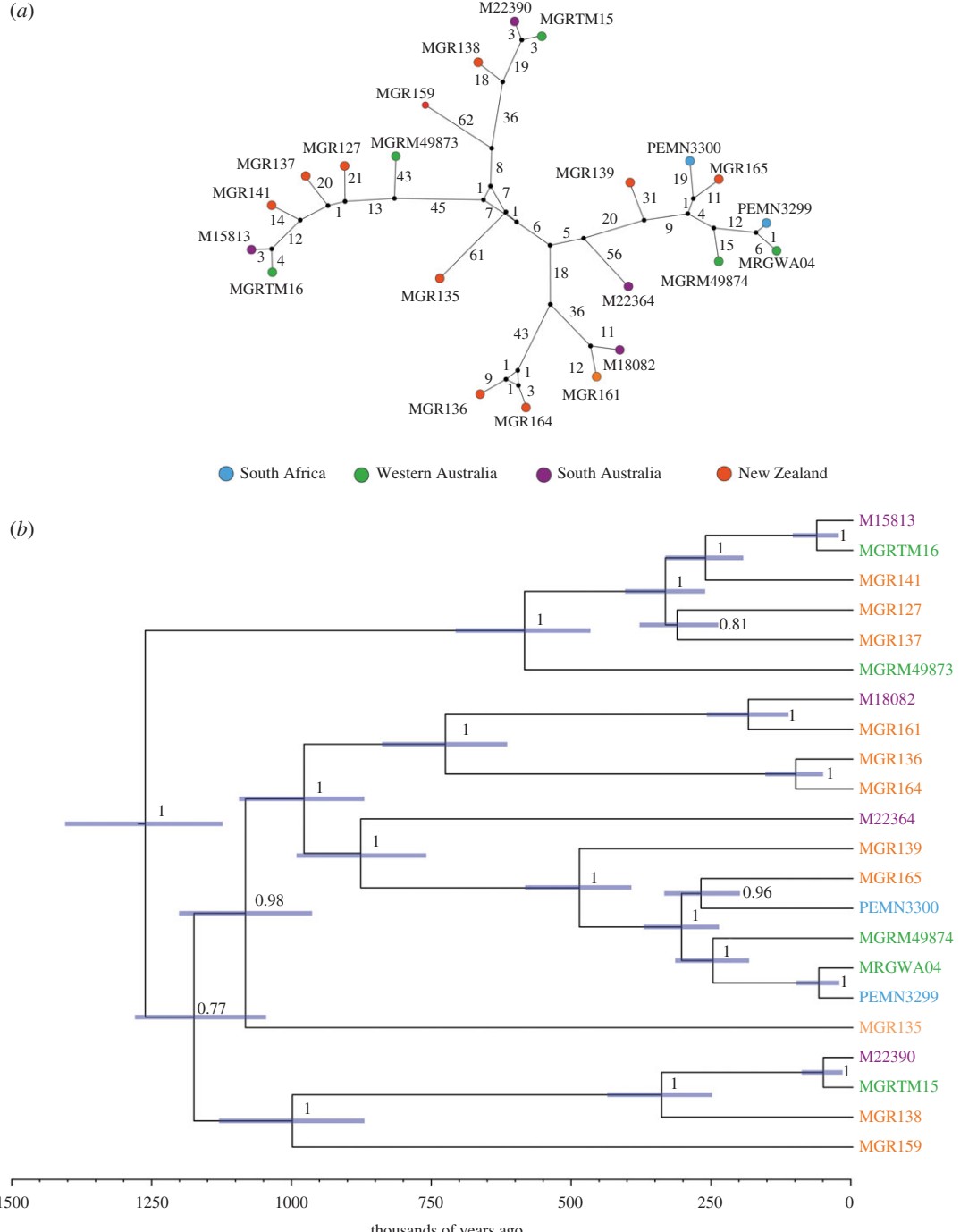

**Figure 3.** The absence of geographic population structure in 22 Gray's beaked whale mitochondrial genomes. (*a*) Haplotype network of mitochondrial genomes. Nodes represent haplotypes, and colours correspond to sampling localities; black nodes represent theoretical, unsampled haplotypes; numbers adjacent to lines indicate the number of mutations between haplotypes. Distances between haplotypes are not to scale. (*b*) Dated mitochondrial phylogeny with node age error bars shown as blue indicating 95% highest posterior density for each divergence, and numbers show posterior probability values.

to be in linkage disequilibrium was considered (electronic supplementary material, figures S1 and S2). Moreover, to investigate for population structure without any *a priori* restrictions on number of populations, we also computed a genome-wide distance matrix and neighbour-joining tree using all individuals mapped to Sowerby's beaked whale. The resultant tree showed no obvious phylogeographic population structure (figure 5*a*). In addition, the fixation index ($F_{ST}$) between individuals grouped by geographical origins did not show any significant differences from 0 (figure 5*b*).

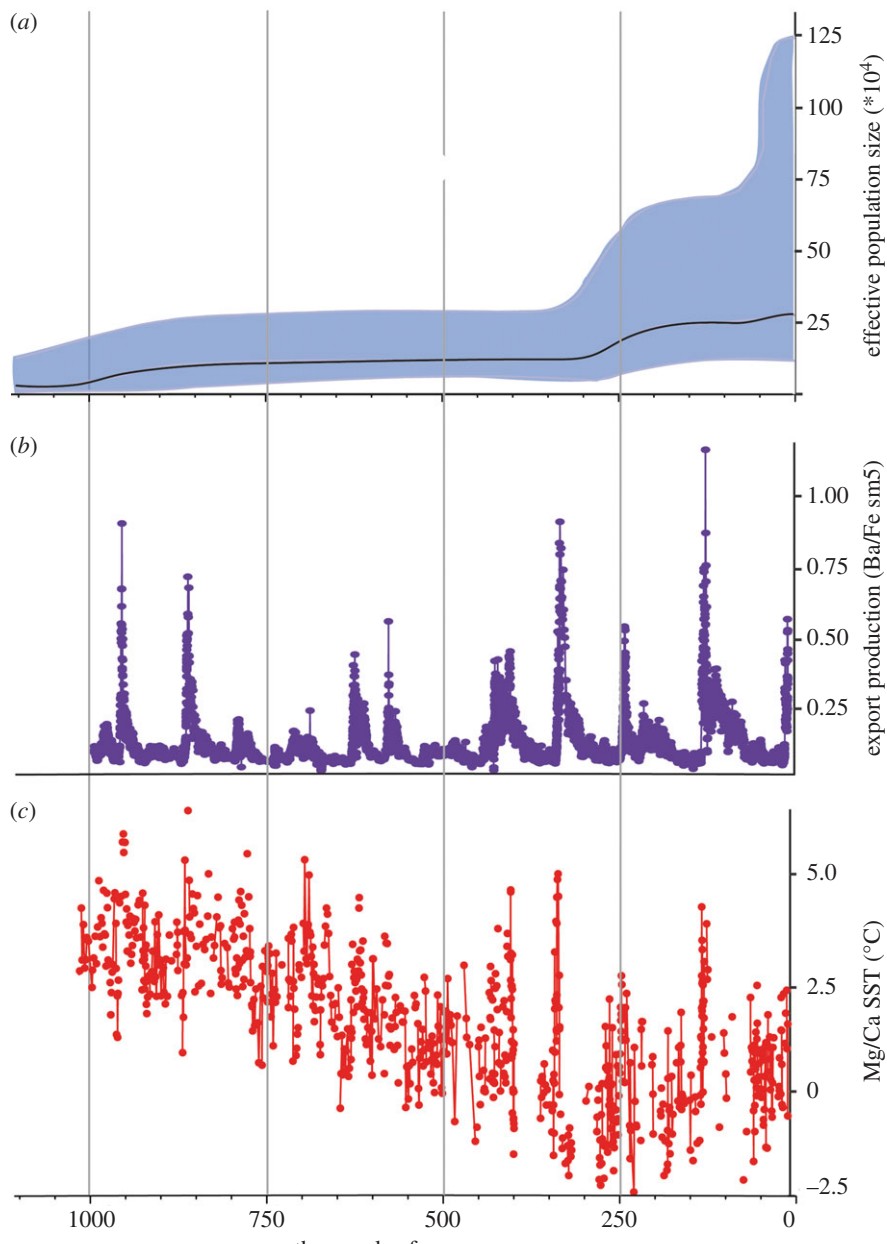

**Figure 4.** Matrilineal demographic history of 22 Gray's beaked whales based on mitochondrial genomes and environmental context. (*a*) Bayesian skyline plot showing median female effective population size and 95% confidence intervals through time; (*b*) export production as described by marine barite (Ba/Fe) as a qualitative proxy across time [55] and (*c*) Mg/Ca in *Noctiluca pachyderma* as a quantitative estimate of past sea surface temperature (SST) [56]. Current SST in the Southern Ocean sampling locations is −2.5°C.

To further investigate the presence of population structure, we used the D-statistics analysis to test for population structure, as this has previously been shown to be more sensitive to small genetic differences among populations [51] (electronic supplementary material, figures S4–S7). This analysis did not uncover any significant differences from 0, regardless of whether we tested 'correct' or 'incorrect' topologies. The only significant difference was retrieved when comparing the 'incorrect' topology of samples from Western Australia and South Africa (*p*-value = 0.024) (electronic supplementary material, figure S4). This result means that when computing D-statistics on the predefined topology [[[Western Australia, South Africa], South Africa], outgroup], the two South African samples appeared more closely related to each other than they were to any of the individuals from Western Australia. However, we had only two samples from South Africa, and this was not seen when comparing South Africa to any other sampling localities. Therefore, it may be a product of not having enough individuals for this specific comparison.

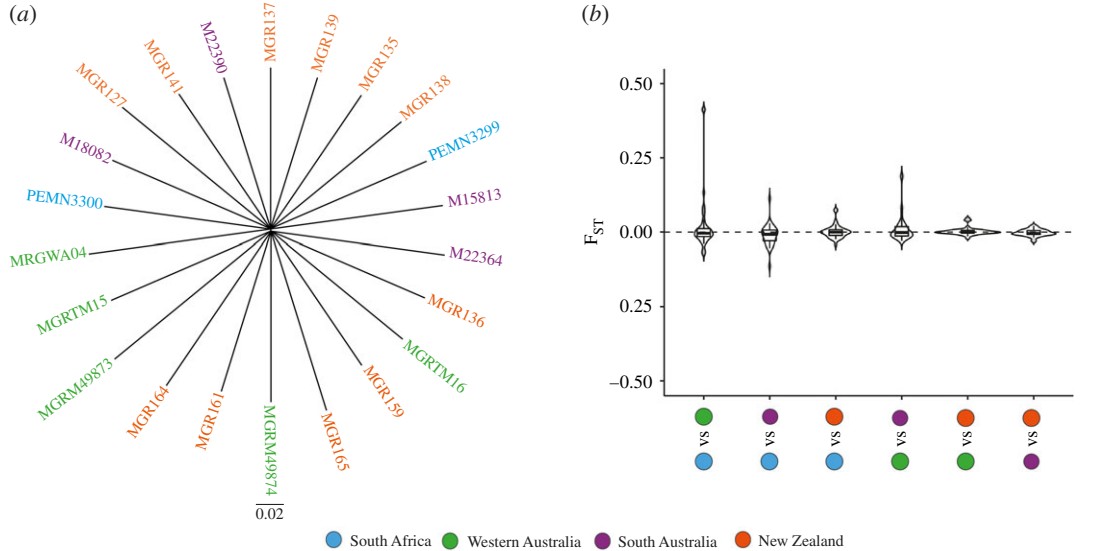

**Figure 5.** The absence of geographic population structure in nuclear genomes. (*a*) Neighbour-joining phylogenetic tree constructed from 3 477 967 segregating sites across autosomes; branch lengths and scale axis show the proportion of site differences between individuals; (*b*) fixation index ($F_{ST}$) between sample localities based on 1 581 396 segregating sites. Variance across the genome was calculated using a window size of 100 kb and a minimum number of sites per window of 500. Significance from 0 was evaluated using a one-sample Wilcoxon signed-rank test in R.

### 3.2.4. Relatedness and sex determination

The highest level of relatedness between two individuals in our dataset was 0.013. This value was recovered from the West Australian individuals MGRTM16 and MGRTM15. Although this is the highest level in our dataset, it is still very low and suggests these individuals are only related to the sixth or seventh degree. A relatedness of one indicates monozygotic twins while 0.5 indicates parent–offspring or siblings.

To determine the sex of the Gray's beaked whales in this study, we used the ratio of the mean coverage on the scaffolds putatively originating from the X chromosome compared to autosomal scaffolds. Despite some differences in the recovered ratios based on mapping reference, the assigned sexes do not differ (electronic supplementary material, tables S5 and S6). The results are concordant with previous methods based on the amplification of the SRY gene for a subsample of the samples (electronic supplementary material, tables S5 and S6) [20]. Concordance between methods shows that low-effort shotgun sequencing is an effective method to assess the sex of Gray's beaked whale individuals. In total, we found 12 males and 10 females.

## 4. Discussion

By performing shallow shotgun sequencing of 22 Gray's beaked whale individuals sampled across approximately 13 000 km of the species' range, we were able to successfully recover complete mitochondrial genomes and partial nuclear genomes from all individuals. This increases the amount of genetic data available for this species, increasing the power of evolutionary inference we can make even with few available samples [57]. Furthermore, by including individuals from new sampling localities in South Africa and South Australia, we are able to broaden inferences to the wider Southern Hemisphere, rather than Australasia specifically.

We recovered no discernible geographic population structure in either the mitochondrial or nuclear genome data, suggesting a lack of both maternal and paternal population structure across the study area in this potentially highly mobile marine predator (figures 3 and 5, electronic supplementary material, figures S1–S7). We acknowledge that our data were derived from stranded animals, and as such we assigned each whale a broad geographic location. However, we do not expect that dead whales would drift across regions. Furthermore, based on the fact that we have so many genome-wide markers, if there was subtle structure, we would expect some degree of genetic clustering between sampling localities.

The absence of clear maternal and biparental population structure across the Southern Hemisphere, as well as a lack of closely related individuals being found at the same site, suggests that both sexes disperse from their natal group, perhaps forming affiliations with unrelated individuals over vast distances. Such a dispersal method has previously been suggested based on genetic kinship data from Gray's beaked whale individuals found in stranded groups [21]. High rates of dispersal, a flexible social system in combination with the rich oceanic habitat of the Southern Hemisphere, could facilitate gene flow and potentially support a surprisingly large number of Gray's beaked whales. The seemingly range-wide gene flow suggested by our data could be facilitated by broad-scale oceanographic features, such as the subtropical convergence that supports large populations of squid [58], and which runs through the centre of Gray's beaked whale distribution (figure 1). High levels of gene flow have also been noted in several Antarctic and Southern Ocean species: highly vagile snow petrels (Pagodroma nivea) [59] and central-place foragers such as penguins (several species) [60]. By contrast, other Southern Hemisphere whales, such as the wide-ranging blue whale (Balaenoptera musculus), do exhibit moderate genetic structure across its range, even though large-scale movements have been reported (up to 6650 km) [61].

Comparisons of mitochondrial genome diversity between Gray's beaked whale and several other cetacean species revealed relatively high levels of maternal genetic diversity within Gray's beaked whale, consistent with previous results using both partial mitochondrial and microsatellite markers (figure 2) [20]. While the same study also indicated a lack of population structure [20], inferences on wider patterns of population structure were not possible as sampling was restricted to two localities only. Here, we further explore the wider population structure in Gray's beaked whales by including whales from South Africa and South Australia. We find that, despite a distance of up to approximately 13 000 km between sampling locations, Gray's beaked whales recovered in South Africa could not be differentiated from the Australasian individuals at either the mitochondrial or nuclear genomic level (figures 3 and 5, electronic supplementary material, figures S1–S7).

To explore the genetic diversity of Gray's beaked whales through time, we performed demographic history analysis on the mitochondrial genomes. We uncover relatively stable and high levels of female effective population size over at least the past approximately 1.1 million years, but an approximate doubling of $N_{ef}$ approximately 250 kya [20] (figure 4a).

The increase in $N_{ef}$ coincides with an interglacial period during a proposed increase in Southern Ocean productivity [55]. This increased productivity is shown as export production, inferred by the presence of marine barite in sediments (Ba/Fe ratio) used as a qualitative proxy [55,62] (figure 4b). During this time, sea surface temperature (SST) [56] also showed a marked increase (figure 4c). Moreover, the peak in sea surface temperature 250 kya, shows the ocean being warmer or at least as warm as it is today. A plausible explanation for the increase in $N_{ef}$ could be that productivity increased in polar waters at the same time as a shift in temperature, which may have allowed an expansion in suitable habitat at this time. Increases in primary production are likely to have provided more food for whale populations, resulting in increased reproduction with subsequent expansion and divergence of existing lineages.

Aside from the doubling in $N_{ef}$ approximately 250 kya, the long-term stability of $N_{ef}$ of the Gray's beaked whale could reflect that its circumpolar habitat has been relatively stable over the long term. Oceanographic features, such as the Antarctic circumpolar current, subtropical convergence and Antarctic polar front (shown in figure 1), have all existed since at least 32–30 Ma, and the land masses of the Southern Ocean have not significantly altered their position for at least 28 Ma [63,64]. However, during periods of recent climate change, for example, the Pleistocene/Holocene transition of approximately 11.7 kya, we do not see any significant changes in maternal $N_{ef}$ (figure 4a).

Our results indicate that Gray's beaked whales have existed as a diverse and stable population across their distribution over the long term, accruing relatively high genetic diversity in comparison with other toothed whales. It is likely that a flexible social system, rich Southern Hemisphere feeding grounds and permanent oceanographic features facilitate gene flow in this species. Given that these whales have never been the focus of whaling—whalers named them 'scamperdown whales' due to their elusive behaviour—Gray's beaked whales appear to have been able to maintain ancestral levels of high genetic diversity and abundance.

In conclusion, through generating complete mitochondrial genomes, a new low-coverage reference panel of nuclear genomes, and a newly scaffolded Sowerby's beaked whale reference genome, we contribute new and complementary insights into the ecology of a relatively unknown whale. We show that Gray's beaked whales have relatively high levels of maternal genetic diversity, which they have retained over the long term, putatively due to the ability of the species to move across oceans. Such

characteristics may facilitate the future survival of this species in a habitat that is undergoing rapid change. Our study shows the ecological information gained from genetic analyses of beached specimens, even with relatively low sequencing effort, and suggests this is a valuable approach for studying highly elusive species where samples are available.

Data accessibility. Genbank accession codes for mitogenomes are MW645443-MW645463 and raw reads for nuclear genomes can be found in Genbank BioProject ID PRJNA702760. Additional Data and relevant code for this research work are stored in GitHub: https://github.com/Mvwestbury/Dstats-topology-test and have been archived within the Zenodo repository: https://doi.org/10.5281/zenodo.4320997

Authors' contributions. K.F.T. conceived the project; M.V.W. and K.F.T. performed the laboratory work; M.V.W., K.F.T., M.L., A.A.C., M.S. and J.A.S.C. performed bioinformatic analyses. R.C. provided samples. K.F.T., E.D.L. and J.R.S. provided funding. E.D.L. provided supervision. M.V.W., K.F.T. and E.D.L. wrote the manuscript with input from all other authors.

Competing interests. We declare we have no competing interests.

Funding. The work was funded by the Lerner Grey Memorial Fund for Marine Research of the American Natural History Museum grant and a University of Exeter European Network Fund (#ENF15) grant to K.F.T. The work was also supported by the Villum Fonden Young Investigator Programme, grant no. 13151, the Carlsberg Foundation Distinguished Associate Professor Fellowship, grant no CF16-0202 and the Independent Research Fund Denmark | Natural Sciences, Forskningsprojekt 1, grant no. 8021-00218B to E.D.L. A.A.C. was funded by the Rubicon-NWO grant (project 019.183EN.005).

Acknowledgements. We thank the following institutions and staff who provided samples for this study: New Zealand Cetacean Tissue Archive—University of Auckland (UoA), South Australia Museum (SAM), West Australia Museum and Port Elizabeth Museum (PEM). Specifically, we thank C. Kemper, S. South, S. Donnellan and D. Stemmer from SAM and G. Hofmeyr from PEM. West Australian samples were collected by N. Gales, R. O'Shea and J. Bannister. New Zealand samples were collected by the Department of Conservation staff and local Māori. Thanks also to S. Jaccard for help with interpretation of Southern Ocean environmental data and C. S. Baker and D. Steel for management of the UoA samples.

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
