## [Peer Review File · Royal Society Open Science]

Review History

RSOS-201788.R0 (Original submission)

Review form: Reviewer 1

Is the manuscript scientifically sound in its present form?

No

Are the interpretations and conclusions justified by the results?

Yes

Is the language acceptable?

Yes

Do you have any ethical concerns with this paper?

No

Have you any concerns about statistical analyses in this paper?

No

Recommendation?

Major revision is needed (please make suggestions in comments)

Comments to the Author(s)

This paper revisits Thompson et al 2016's examination of the population structure of grey's beaked whale using fewer samples but more genomic markers and whole mitochondrial genomes. The paper is reasonably well-written, but needs to be clearer about the novelty of the findings presented vs what was previously published. I also have concerns and suggestions about the analyses.

The previous study found no detectable genetic population structure using 94 samples, 530 bp of mtDNA control region and 12 microsats. This work builds on that paper with 2 samples from South African and 4 from South Australia. The natural extension of this work would be to see if the lack of population structure holds with more markers, as simulation work suggests that more markers rather than more samples are better at detecting subtle population structure. Indeed, while many marine mammals do show population structure, often linked to behavioural or social factors, many marine species have high effective population sizes and low levels of divergence, including in sharks and fish. It would be good to consider these in the context of the work, as methods that test for subtle population structure are probably of more relevance to many readers.

Major comments

Abstract: needs to highlight more clearly what samples are new and what have been analysed previously, likewise findings. For example, high levels of mtDNA diversity, no population structure and stable demographic history was found in Thompson et al 2016.

Nuclear analyses:

How was linkage taken into account? For example, Foote et al 2019 restricted the analyses to sites greater than ≥ 20 kb apart to avoid linkage when doing PCA. Perhaps I missed something as I did not read the sup mat closely but linked loci are not independent and so should not be treated as separate markers for PCA or genetic clustering algorithms.

Genomic markers:

How genotyping was conducted and how many markers were used in subsequent analyses needs to be clearer in the main body of the manuscript. Presumably some threshold/confidence on genotype likelihood was used? How many loci were used for each of the specific analyses? What was the average depth for loci, and number of individuals typed at each loci, and did this vary between analyses?

Overall, the analyses were standard but not specifically tailored to detecting low levels of divergence, if any existed.

- Was F_{ST} calculated between sample sites, e.g., NZ, SA ?
- Was DAPC considered? It is more sensitive at lower levels of divergence than PCA
- Was outlier analysis considered? Loci under selection are often more sensitive to population structure than neutral markers
- How did patterns of relatedness vary across sites? Given the jump from 12 to some unspecified by presumably larger number of nuclear markers will give a much better impression of relatedness.

Given the large number of genetic markers available, it seems that more analyses could be done to answer this specific question - one that was already obvious from the results of Thompson et al 2016.

D-statistic analyses

Line 168: I've not seen this approach before, a reference to where it's been done before would be useful.

What was the outgroup used in this analysis? How many loci were used? Given the small sample sizes, is this analysis really contributing much. Also South Australia is part of Australasia (line 253) so there are only 2 samples from outside the distribution previously analysed (ie, 2 from South Africa).

Sex determination:

Not sure what this section contributes to the paper? It's not mentioned in the Methods or Discussion. The NZ and WA samples were sexed in Thompson et al 2016 so it seems that the other samples might have been similarly sex determined by PCR as well? As sex is not used in any analyses that I can see think this could go in the supplementary material.

Discussion: This work builds on the previous Heredity article. It should be clearly stated what is novel and what confirms previous analyses, including Ne estimates and Bayesian skyline plot findings.

Figure 1: Nice map, but please add sample sizes, and I'm not sure what panel c adds to this?

Minor comments

Line 55: actually, sites like Hawaii, Bahamas and Canary Islands have provided substantial information on Cuvier's and Blainville's ecology.

Paragraph line 92: include new and previously analysed sample sizes in this paragraph

Line 144: killer whale is the common name of *Orcinus orca*

Line 236: table X ?

Review form: Reviewer 2

Is the manuscript scientifically sound in its present form?

No

Are the interpretations and conclusions justified by the results?

No

Is the language acceptable?

Yes

Do you have any ethical concerns with this paper?

No

Have you any concerns about statistical analyses in this paper?

Yes

Recommendation?

Accept with minor revision (please list in comments)

Comments to the Author(s)

Westbury et al. have sequenced mitogenomes and partial nuclear genomes to elucidate the phylogeographic and demographic history of Gray's beaked whale. This is a valuable contribution to the literature that backs up previous hypotheses based off partial mitochondrial control region and microsatellite data. I have only minor comments:

In the methods can you please bring in some of the supplementary information to justify decisions and make results clear, namely why the killer whale genome was used as a scaffold, and not say sperm whale; and brief methodology on DNA sexing, as it jumps out in the results.

In the DNA sexing section of the results can you please state that you are sexing Gray's beaked whale as it is not clear.

For the demographic reconstructions using whole mitogenomes, can you please repeat your analyses with a constant population size prior, and compare this to the Bayesian Skyline prior (shown in the paper) using model comparison stats (e.g. Bayes Factor), to determine whether a constant population size or dynamic population size is supported.

Decision letter (RSOS-201788.R0)

Dear Ms Thompson

The Editors assigned to your paper RSOS-201788 "Ocean-wide genomic variation in Gray's beaked whales, *Mesoplodon grayi*" have now received comments from reviewers and would like you to revise the paper in accordance with the reviewer comments and any comments from the Editors. Please note this decision does not guarantee eventual acceptance.

Please submit your revised manuscript and required files (see below) no later than 21 days from today's (ie 16-Nov-2020) date. Note: the ScholarOne system will 'lock' if submission of the revision is attempted 21 or more days after the deadline. If you do not think you will be able to meet this deadline please contact the editorial office immediately.

Please note article processing charges apply to papers accepted for publication in Royal Society Open Science (<https://royalsocietypublishing.org/rsos/charges>). Charges will also apply to papers transferred to the journal from other Royal Society Publishing journals, as well as papers submitted as part of our collaboration with the Royal Society of Chemistry

(<https://royalsocietypublishing.org/rsos/chemistry>). Fee waivers are available but must be requested when you submit your revision (<https://royalsocietypublishing.org/rsos/waivers>).

on behalf of Prof Pete Smith (Subject Editor)
openscience@royalsociety.org

Associate Editor Comments to Author:

Two reviewers have provided a number of queries, comments and suggestions. Please ensure you fully respond to these in your revision and point-by-point response document.

Reviewer comments to Author:

Reviewer: 1

Comments to the Author(s)

This paper revisits Thompson et al 2016's examination of the population structure of grey's beaked whale using fewer samples but more genomic markers and whole mitochondrial genomes. The paper is reasonably well-written, but needs to be clearer about the novelty of the findings presented vs what was previously published. I also have concerns and suggestions about the analyses.

The previous study found no detectable genetic population structure using 94 samples, 530 bp of mtDNA control region and 12 microsats. This work builds on that paper with 2 samples from South African and 4 from South Australia. The natural extension of this work would be to see if the lack of population structure holds with more markers, as simulation work suggests that more markers rather than more samples are better at detecting subtle population structure. Indeed, while many marine mammals do show population structure, often linked to behavioural or social factors, many marine species have high effective population sizes and low levels of divergence, including in sharks and fish. It would be good to consider these in the context of the work, as methods that test for subtle population structure are probably of more relevance to many readers.

Major comments

Abstract: needs to highlight more clearly what samples are new and what have been analysed previously, likewise findings. For example, high levels of mtDNA diversity, no population structure and stable demographic history was found in Thompson et al 2016.

Nuclear analyses:

How was linkage taken into account? For example, Foote et al 2019 restricted the analyses to sites greater than ≥ 20 kb apart to avoid linkage when doing PCA. Perhaps I missed something as I did not read the sup mat closely but linked loci are not independent and so should not be treated as separate markers for PCA or genetic clustering algorithms.

Genomic markers:

How genotyping was conducted and how many markers were used in subsequent analyses needs to be clearer in the main body of the manuscript. Presumably some threshold/confidence on genotype likelihood was used? How many loci were used for each of the specific analyses? What was the average depth for loci, and number of individuals typed at each loci, and did this vary between analyses?

Overall, the analyses were standard but not specifically tailored to detecting low levels of divergence, if any existed.

- Was FST calculated between sample sites, e.g., NZ, SA ?
- Was DAPC considered? It is more sensitive at lower levels of divergence than PCA
- Was outlier analysis considered? Loci under selection are often more sensitive to population structure than neutral markers
- How did patterns of relatedness vary across sites? Given the jump from 12 to some unspecified by presumably larger number of nuclear markers will give a much better impression of relatedness.

Given the large number of genetic markers available, it seems that more analyses could be done to answer this specific question - one that was already obvious from the results of Thompson et al 2016.

D-statistic analyses

Line 168: I've not seen this approach before, a reference to where it's been done before would be useful.

What was the outgroup used in this analysis? How many loci were used? Given the small sample sizes, is this analysis really contributing much. Also South Australia is part of Australasia (line 253) so there are only 2 samples from outside the distribution previously analysed (ie, 2 from South Africa).

Sex determination:

Not sure what this section contributes to the paper? It's not mentioned in the Methods or Discussion. The NZ and WA samples were sexed in Thompson et al 2016 so it seems that the other samples might have been similarly sex determined by PCR as well? As sex is not used in any analyses that I can see think this could go in the supplementary material.

Discussion: This work builds on the previous Heredity article. It should be clearly stated what is novel and what confirms previous analyses, including Ne estimates and Bayesian skyline plot findings.

Figure 1: Nice map, but please add sample sizes, and I'm not sure what panel c adds to this?

Minor comments

Line 55: actually, sites like Hawaii, Bahamas and Canary Islands have provided substantial information on Cuvier's and Blainville's ecology.

Paragraph line 92: include new and previously analysed sample sizes in this paragraph

Line 144: killer whale is the common name of *Orcinus orca*

Line 236: table X ?

Reviewer: 2

Comments to the Author(s)

Westbury et al. have sequenced mitogenomes and partial nuclear genomes to elucidate the phylogeographic and demographic history of Gray's beaked whale. This is a valuable contribution to the literature that backs up previous hypotheses based off partial mitochondrial control region and microsatellite data. I have only minor comments:

In the methods can you please bring in some of the supplementary information to justify decisions and make results clear, namely why the killer whale genome was used as a scaffold, and not say sperm whale; and brief methodology on DNA sexing, as it jumps out in the results.

In the DNA sexing section of the results can you please state that you are sexing Gray's beaked whale as it is not clear.

For the demographic reconstructions using whole mitogenomes, can you please repeat your analyses with a constant population size prior, and compare this to the Bayesian Skyline prior (shown in the paper) using model comparison stats (e.g. Bayes Factor), to determine whether a constant population size or dynamic population size is supported.

===PREPARING YOUR MANUSCRIPT===

===PREPARING YOUR REVISION IN SCHOLARONE===

To revise your manuscript, log into <https://mc.manuscriptcentral.com/rsos> and enter your Author Centre - this may be accessed by clicking on "Author" in the dark toolbar at the top of the

page (just below the journal name). You will find your manuscript listed under "Manuscripts with Decisions". Under "Actions", click on "Create a Revision".

Author's Response to Decision Letter for (RSOS-201788.R0)

See Appendix A.

RSOS-201788.R1 (Revision)

Review form: Reviewer 1

Is the manuscript scientifically sound in its present form?

Yes

Are the interpretations and conclusions justified by the results?

Yes

Is the language acceptable?

Yes

Do you have any ethical concerns with this paper?

No

Have you any concerns about statistical analyses in this paper?

No

Recommendation?

Accept as is

Comments to the Author(s)

Thank you for addressing the reviewer's concerns and congratulations on an impressive swathe of genomic analyses.

Review form: Reviewer 2

Is the manuscript scientifically sound in its present form?

Yes

Are the interpretations and conclusions justified by the results?

Yes

Is the language acceptable?

Yes

Do you have any ethical concerns with this paper?

No

Have you any concerns about statistical analyses in this paper?

No

Recommendation?

Accept as is

Comments to the Author(s)

Westbury et al. have done a thorough and excellent job in revising their manuscript. I am satisfied with the changes and new analyses that have been incorporated. I have no further comments.

Decision letter (RSOS-201788.R1)

Dear Ms Thompson,

It is a pleasure to accept your manuscript entitled "Ocean-wide genomic variation in Gray's beaked whales, *Mesoplodon grayi*" in its current form for publication in Royal Society Open Science. The comments of the reviewers who reviewed your manuscript are included at the foot of this letter.

Please now deposit raw seq data and genomic sequences to NCBI before we proceed to the production stage. For papers describing genome assemblies, RNA sequencing, or gene expression estimates we require the following:

- All raw RNA seq data should be deposited and made available in the NCBI Sequence Read Archive: <https://www.ncbi.nlm.nih.gov/sra/docs/submit/>
- Genome assemblies and nucleotide data should be deposited within GenBank: <https://www.ncbi.nlm.nih.gov/genbank/>

Once you have the GenBank accession IDs, please reply to this email with the appropriate accessions to be added to your data accessibility statement in ScholarOne.

After this, you can expect to receive a proof of your article in the near future. Please contact the editorial office (openscience@royalsociety.org) and the production office (openscience_proofs@royalsociety.org) to let us know if you are likely to be away from e-mail contact – if you are going to be away, please nominate a co-author (if available) to manage the proofing process, and ensure they are copied into your email to the journal.

on behalf of Professor Pete Smith (Subject Editor)
openscience@royalsociety.org

Associate Editor Comments to Author:

Please accept our apologies for the delay in reviewing - one of the reviewers was unexpectedly delayed by an injury (and the editors are very grateful to them for nevertheless completing the report with this additional difficulty). In any case, the general view is that your revisions have resolved the concerns the referees had. Please ensure that you make your data publicly accessible on receipt of this message and contact the editorial office to confirm.

Reviewer comments to Author:

Reviewer: 1

Comments to the Author(s)

Thank you for addressing the reviewer's concerns and congratulations on an impressive swathe of genomic analyses.

Reviewer: 2

Comments to the Author(s)

Westbury et al. have done a thorough and excellent job in revising their manuscript. I am satisfied with the changes and new analyses that have been incorporated. I have no further comments.

Appendix A

Response to reviewers and revision notes for Manuscript ID RSOS-201788 entitled 'Ocean-wide genomic variation in Gray's beaked whales, *Mesoplodon grayi*'

We thank the reviewers for their insightful comments, please see responses below:

Reviewer: 1

Comments to the Author(s)

This paper revisits Thompson et al 2016's examination of the population structure of grey's beaked whale using fewer samples but more genomic markers and whole mitochondrial genomes. The paper is reasonably well-written, but needs to be clearer about the novelty of the findings presented vs what was previously published. I also have concerns and suggestions about the analyses.

The previous study found no detectable genetic population structure using 94 samples, 530 bp of mtDNA control region and 12 microsats. This work builds on that paper with 2 samples from South African and 4 from South Australia. The natural extension of this work would be to see if the lack of population structure holds with more markers, as simulation work suggests that more markers rather than more samples are better at detecting subtle population structure. Indeed, while many marine mammals do show population structure, often linked to behavioural or social factors, many marine species have high effective population sizes and low levels of divergence, including in sharks and fish. It would be good to consider these in the context of the work, as methods that test for subtle population structure are probably of more relevance to many readers.

Answer: Many thanks for this comment, please find changes throughout the text shown in red to highlight the additional samples used in the current study.

Major comments

Q1. Abstract: needs to highlight more clearly what samples are new and what have been analysed previously, likewise findings. For example, high levels of mtDNA diversity, no population structure and stable demographic history was found in Thompson et al 2016.

Answer: Please see additional text and rewording of the Abstract shown in red, Specifically, please see changes on lines 25–29 and lines 31–33. Also please see, lines 100–104 and 113–114 in the Introduction.

Q2. Nuclear analyses:

How was linkage taken into account? For example, Foote et al 2019 restricted the analyses to sites greater than ≥ 20 kb apart to avoid linkage when doing PCA. Perhaps I missed something as I did not read the sup mat closely but linked loci are not independent and so should not be treated as separate markers for PCA or genetic clustering algorithms.

Answer: We have now also filtered the dataset for sites under linkage disequilibrium using the software ngsLD. We have rerun the PCA and admixture proportion analyses with this

subset of sites and see that although there are some minor differences, the overall lack of population structure remains (supplementary fig S3).

Q3. Genomic markers:

How genotyping was conducted and how many markers were used in subsequent analyses needs to be clearer in the main body of the manuscript. Presumably some threshold/confidence on genotype likelihood was used? How many loci were used for each of the specific analyses? What was the average depth for loci, and number of individuals typed at each loci, and did this vary between analyses?

Answer: We have now moved the majority of the methods from the supplements into the main text, seen in red. Hopefully it is now clearer that we did not use genotyping but instead genotype likelihoods which does not require a threshold/confidence as the likelihood of all bases at a given site are considered. As the analyses we performed are specifically made for low coverage data we did not require a minimum depth but did specify that the site needs to be found in at least 50% of the individuals. We kept all parameters consistent between analyses where possible unless stated otherwise. Finally, we have included the number of sites for each analysis in the figure legends of each analysis.

Q4. Overall, the analyses were standard but not specifically tailored to detecting low levels of divergence, if any existed.

- Was FST calculated between sample sites, e.g., NZ, SA ?

Answer: We have now included Fst between sites in the manuscript which also support no population structure based on sampling location (Fig 5). Please see methods section, lines 266–277 and results section, lines 380–381 and the additional Figure 5(b) that shows the Fst results.

Q5. Was DAPC considered? It is more sensitive at lower levels of divergence than PCA

Answer: We investigated running this analysis however it appears to not be suitable for low coverage whole genome data so we decided to not include it. In turn, PCAngsd was specifically designed for low coverage data like ours and it uses an iterative process which should hopefully maximise population structure if it is present. Although we did not use DAPC, we believe the plethora of other analyses we performed and the congruence between nuclear and mitochondrial datasets are sufficient to support our proposed lack of population structure in the Grays' beaked whale.

Q6. Was outlier analysis considered? Loci under selection are often more sensitive to population structure than neutral markers

Answer: We did not consider this analyses as we wanted to look at the population structure across the entire dataset to avoid biases that may occur when choosing loci that fit a previously determined hypothesis (e.g. loci that support population structure).

Q7. How did patterns of relatedness vary across sites? Given the jump from 12 to some unspecified by presumably larger number of nuclear markers will give a much better impression of relatedness.

Answer: We have now included relatedness estimates within sites which show none of the individuals from similar geographic origins were related. Please see text lines 305–312 in methods section and lines 394–399 in the results section.

Given the large number of genetic markers available, it seems that more analyses could be done to answer this specific question - one that was already obvious from the results of Thompson et al 2016.

Q8. D-statistic analyses

Line 168: I've not seen this approach before, a reference to where it's been done before would be useful.

What was the outgroup used in this analysis? How many loci were used? Given the small sample sizes, is this analysis really contributing much. Also South Australia is part of Australasia (line 253) so there are only 2 samples from outside the distribution previously analysed (ie, 2 from South Africa).

Answer: This approach has been used a number of times but is not extensive in the literature although we have included some example references in the main text. We also included a more detailed description of the reasoning behind the method to help guide readers unfamiliar with the method and moved the parameters as well as the outgroup used from the supplementary information to the main text. The total number of loci used in this analysis is not as relevant as the window number which was 2,859 1MB windows. Within these windows the number of sites considered (i.e. ABBA + BABA sites) ranged from 20,012 - 130,238 with a mean of 65,569 and standard deviation of 19,564. This analysis is contributing to our broader study as it is not being considered on its own but rather adding additional information and confirming the nuclear results proposed by the PCA, admixture proportions, phylogenetic tree (now also the Fst).

Q9. Sex determination:

Not sure what this section contributes to the paper? It's not mentioned in the Methods or Discussion. The NZ and WA samples were sexed in Thompson et al 2016 so it seems that the other samples might have been similarly sex determined by PCR as well? As sex is not used in any analyses that I can see think this could go in the supplementary material.

Answer: We provide a description of the method used to sex the new and previously analysed samples as a suggestion for other studies that may focus on low-coverage genomics. All results were consistent between this and previous data, providing a rapid and effective method to derive sex identification in similar studies.

Q10. Discussion: This work builds on the previous Heredity article. It should be clearly stated what is novel and what confirms previous analyses, including Ne estimates and Bayesian skyline plot findings.

Answer: We have added several sentences within the paragraph reporting the results of the demographic history analyses that compare the two studies. Please see lines 348–354 for the amended text.

Q11. Figure 1: Nice map, but please add sample sizes, and I'm not sure what panel c adds to this?

Answer: We thank you for your comments and have altered the map in Figure 1 as suggested.

Minor comments

Q12. Line 55: actually, sites like Hawaii, Bahamas and Canary Islands have provided substantial information on Cuvier's and Blainville's ecology.

Answer: Although this information is known for some species of beaked whale, we wanted to highlight the lack of information in beaked whales as a whole. We added the text “lacking for most species” to extrapolate to a beaked whale wide context.

Q13. Paragraph line 92: include new and previously analysed sample sizes in this paragraph

Answer: Please find altered text shown in red.

Q14. Line 144: killer whale is the common name of *Orcinus orca*

Answer: We thank the reviewer for their comment however, the killer whale is also commonly referred to as the orca, therefore we would prefer to keep the common name as orca.

Q15. Line 236: table X ?

Answer: Many thanks for this comment on a typo, please find the text corrected.

Reviewer: 2

Comments to the Author(s)

Westbury et al. have sequenced mitogenomes and partial nuclear genomes to elucidate the phylogeographic and demographic history of Gray's beaked whale. This is a valuable contribution to the literature that backs up previous hypotheses based off partial mitochondrial control region and microsatellite data. I have only minor comments:

Q1. In the methods can you please bring in some of the supplementary information to justify decisions and make results clear, namely why the killer whale genome was used as a scaffold, and not say sperm whale; and brief methodology on DNA sexing, as it jumps out in the results.

Answer: We have now moved the more detailed supplementary methods into the main text. We selected the orca genome as it has a high-quality genome assembly and the most closely related species with such an assembly. However, we do note that the orca is equally divergent

from beaked whales as all other species within Delphinoidea, a number of which also have high quality genome assemblies so would have been just as suitable (for example, the beluga and bottlenose dolphin). The sperm whale does have a chromosome level assembly but is more divergent which may have resulted in even more inflated reference biases or less efficient mapping.

Q2. In the DNA sexing section of the results can you please state that you are sexing Gray's beaked whale as it is not clear.

Answer: Please find that the text has been altered as suggested. Please see changes lines 313–329.

Q3. For the demographic reconstructions using whole mitogenomes, can you please repeat your analyses with a constant population size prior, and compare this to the Bayesian Skyline prior (shown in the paper) using model comparison stats (e.g. Bayes Factor), to determine whether a constant population size or dynamic population size is supported.

Answer: We thank the reviewer for their comment and have now completed this analysis as requested using the Nested Sampling analyses in BEAST to compare models. In addition we performed an additional extended BSP analyses to estimate the number of population size changes. Please see amended text lines, 118-119 in the methods section, and lines 340–345 in the results.